# Surface Subsidence Monitoring of Mining Areas in Hunan Province Based on Sentinel-1A and DS-InSAR

**DOI:** 10.3390/s23198146

**Published:** 2023-09-28

**Authors:** Liya Zhang, Pengfei Gao, Zhengzheng Gan, Wenhao Wu, Yafeng Sun, Chuanguang Zhu, Sichun Long, Maoqi Liu, Hui Peng

**Affiliations:** 1School of Earth Science and Spatial Information Engineering, Hunan University of Science and Technology, Xiangtan 411201, China; lyzhang47@163.com (L.Z.); gaopf426@163.com (P.G.); gzz_hnust@163.com (Z.G.); zhuchuanguang@163.com (C.Z.); sclong@hnust.edu.cn (S.L.); mqliu@mail.hnust.edu.cn (M.L.); pphh0824@163.com (H.P.); 2Shanghai Urban Construction Design & Research Institute, Shanghai 200125, China; sunyafeng@sucdri.com

**Keywords:** DS-InSAR, mining subsidence survey, mining railway

## Abstract

Monitoring the surface subsidence in mining areas is conducive to the prevention and control of geological disasters, and the prediction and early warning of accidents. Hunan Province is located in South China. The mineral resource reserves are abundant; however, large and medium-sized mines account for a low proportion of the total, and the concentration of mineral resource distribution is low, meaning that traditional mining monitoring struggles to meet the needs of large-scale monitoring of mining areas in the province. The advantages of Interferometric Synthetic Aperture Radar (InSAR) technology in large-scale deformation monitoring were applied to identify and monitor the surface subsidence of coal mining fields in Hunan Province based on a Sentinel-1A dataset of 86 images taken from 2018 to 2020, and the process of developing surface subsidence was inverted by selecting typical mining areas. The results show that there are 14 places of surface subsidence in the study area, and accidents have occurred in 2 mining areas. In addition, the railway passing through the mining area of Zhouyuan Mountain is affected by the surface subsidence, presenting a potential safety hazard.

## 1. Introduction

Hunan Province is located in the middle reaches of the Yangtze River in China. It is the transitional zone from the Yunnan–Guizhou Plateau to the Jiangnan Hills and from the Nanling Mountains to the Jianghan Plain, and it is rich in mineral resources. There are three major geologically metallogenic tectonic units in Hunan Province: central Hunan, the southeast Hunan fold area, and the Bamian Mountain fold area and Xuefeng Mountain uplift area. These favorable geological structures contribute to good conditions for metallogenesis. However, the continuous mining of underground coal resources is liable to cause damage to the stability of the geological conditions around the mining area, change the local ecological and geological environment of the mining area, and induce geological disasters such as surface subsidence, collapse of mining fields, landslides, debris flow, and so on. Especially in rainy seasons, abundant rainfall provides favorable conditions for the development of geological disasters in mining areas, seriously endangering the safety of residents and greatly impeding the sustainable development of ecology and economy in mining areas. Therefore, the monitoring and identification of surface subsidence in a large range of mining areas, as well as the analysis of its temporal and spatial distribution characteristics, evolution rules, and influenced areas, can offer a scientific basis for local governments to carry out the early warning and prediction of geological disasters, relocation of residents, protection of buildings, restoration and management of the ecological environment, and other work.

At present, surface subsidence monitoring and geological hazard identification in mining areas mainly rely on electronic total station surveys, leveling, and Global Navigation Satellite System (GNSS) technology. However, the work efficiency of these methods is low, barely enough to take into account the monitoring and identification tasks of very few mining fields at the same time, and it cannot feasibly meet the needs of large-scale monitoring and identification of multiple mining areas at the same time. In addition, these traditional monitoring methods can only reflect the subsidence or displacement development of discrete monitoring points, and they are less likely to accurately reflect the full landscape and development characteristics of surface subsidence in mining areas. Hunan Province is rich in mineral resources, but the resources are widely distributed, the proportion of large and medium-sized mines is relatively small, and the concentration of mineral resources is low. As a result, the above methods have become difficult to adapt to the current monitoring needs of large-scale surface deformation, and it is urgent to introduce new technologies that can efficiently implement large-scale monitoring.

Since the late 1960s, measurement techniques and theories have continued to develop, resulting in the development of the Persistent Scatterers InSAR (PS-InSAR) technique [1,2], Small Baseline Subset InSAR (SBAS-InSAR) technology [3], Distributed Scatterers InSAR (DS-InSAR) technology [4], and other sequential InSAR technology. InSAR has the characteristics of high temporal and spatial resolution, wide monitoring range, high monitoring accuracy, low cost, all-day and all-weather observation, and so on [5]. Coupled with the characteristics based on surface observation, InSAR has unparalleled advantages over traditional methods in monitoring surface subsidence in mining areas. At present, InSAR has been widely used in the monitoring of landslides [6,7], volcanic activities [8,9,10], earthquake deformation [11], and mining deformation [12,13,14,15,16].

Many scholars have applied sequential InSAR to the monitoring surface subsidence in China, and they have achieved a wealth of research results, but most of these results focus on the identification of large-scale geological hazards or the urban surface subsidence in North China, where surface subsidence is relatively serious. For example, Zhang et al. [17] extracted the surface subsidence of the Beijing–Tianjin–Hebei region from 1992 to 2014 based on the the multiple master-image coherent target small-baseline InSAR (MCTSB-InSAR) method using ERS-1/2, ENVISAT ASAR, and RADARSAT-2 SAR image datasets of different periods, and for the first time, the InSAR was used to map the land surface subsidence over a long time span and large scale in China. Zhang et al. [18] used a Sentinel-1A SAR image dataset to extract the surface displacement of the Tianjin area from 2016 to 2017 based on Multi-temporal InSAR technology, and evaluated the monitoring results in conjunction with eight GNSS stations. Huang et al. [19] combined Joint-Scatterer InSAR and SBAS-InSAR technologies to identify landslide geological hazards in the Dadu River basin in southwest China, and pointed out that the spatial distribution of landslides is closely related to its geological and climatic characteristics, and the differences between upstream and downstream are significant. Zhang et al. [20] developed a geological hazard monitoring process along the transmission line based on InSAR technology and identified 105 hidden geological hazard points along a transmission line in China. However, in terms of the surface deformation monitoring of coal mines in Hunan Province, China, only a few scholars have applied InSAR technology to the surface deformation monitoring research of mining areas, and they have achieved some research results. Yin et al. [21], based on the SBAS-InSAR technology and data of nine ALOS images, inverted the temporal surface deformation in a tin mining field in Xikuang Mountain, Hunan Province, from January 2007 to January 2009, and discussed the influence areas and development of surface subsidence in detail. Liu et al. [22], based on the SBAS-InSAR technology and data of 24 Sentinel-1A images, extracted the sequential surface deformation variables of a typical rock salt mining field in Changde, Hunan Province, from June 2015 to January 2017. The results show that the surface subsidence in the monitored mining area presents a lagging and seasonal fluctuation. However, these results are only applicable for a small range or a single mining area of surface deformation monitoring. Additionally, this research is slightly outdated and not entirely relevant to our studies.

Based on InSAR technology, we explored the Sentinel-1A SAR image dataset captured during 2018–2020 to extract the surface subsidence information of four groups of mines in Hunan, China, since few studies have been conducted on the identification of surface subsidence of mining areas in Hunan, China, on a large scale, and the deformation rate map of the study area was also drawn. Still, the development process of surface subsidence of four typical mining fields in the study area was inverted, and the impact of the railway across the mining area on the surface subsidence was evaluated.

## 2. Materials and Methods

### 2.1. Study Region

Hunan Province is located between latitudes 24°38${}^{\prime }$ N and 30°08${}^{\prime }$ N and longitudes 108°47${}^{\prime }$ E and 114°15${}^{\prime }$ E. It spans about 774 km from north to south and 667 km from west to east. Its landforms are mainly hilly and mountainous. Hunan Province is rich in mineral resources with superior conditions of metallogenesis, but the distribution of resources is uneven and the concentration rate of mine distribution is low. Hunan has a subtropical monsoon climate, with abundant light and heat, rain and heat in the same period, and abundant rainfall. The rainy season is mainly from late March to mid-July each year, which is the peak season for the occurrence of geological disasters in the province. In addition, superior light and heat and abundant rainfall reserves are conducive to the growth of forests, vegetation, and crops, which leads to serious decoherence in the deformation monitoring process using InSAR technology. Underground coal mining activities are active in Hunan, and resource exploitation is fairly common. In 2022, 13 safety accidents were reported due to unreasonable or excessive mining, which not only caused irreparable economic losses, but also endangered the life of miners or local residents. However, the surface subsidence caused by mining is still unclear in many coal mining areas in Hunan. Thereby, this study takes four groups of mines (named A–D) with a relatively concentrated distribution of coal mining in Hunan Province as the research area to carry out large-scale surface deformation monitoring, as shown in Figure 1.

### 2.2. Data

The study area in this paper is completely covered by the image data of the Sentinel-1A ascent orbit with VV polarization IW mode in 4 different frames of orbit 11 and orbit 113 (Table 1). The image dataset of the Sentinel-1A ascent orbit taken from January 2018 to December 2020 was selected to carry out long-term and large-scale surface deformation monitoring on four groups of mines in Hunan Province. For each frame, 31 images were selected for DS-InSAR processing, and the spatiotemporal baseline information of the image data is shown in Figure 2. The 90 m resolution TanDEM-X90 DEM data published in 2018 are more superior than the SRTM data published earlier. Therefore, in order to better eliminate the terrain phase, TanDEM-X90 DEM data were used for the compensation of the terrain phase and geocoding. Finally, we collected the specific locations of each group of mines and their valid time for mining rights, so as to judge whether the mining fields were mined in advance and violated the law, by comparing with the DS-InSAR monitoring results. The specific coverage of images, distribution of mining areas, and topographic relief in the study area are shown in Figure 1.The rainy season in the study area is mainly concentrated from late March to mid-July every year, and sufficient rainfall in the rainy season is also conducive to the growth of vegetation. As a result, the vegetation coverage can cause incoherent noise. Meanwhile, atmospheric environment differences can lead to densely atmospheric streaks in the differential interferogram dataset. To address these problems, Sentinel-1A SAR image data captured on 1 January 2019, were selected as the main reference images, and other images were slave images to generate a differential interferogram of a time sequence.

### 2.3. Method

The high coverage of vegetation in the study area will lead to the intensification of the decoherence effect and further affect the observation results of InSAR. In addition, most of the mining areas are mainly distributed at the edge of the villages or mountainous areas, and the distribution of permanent scatterers in the mining areas is relatively rare. Therefore, in order to avoid the effects caused by all these adverse factors, DS-InSAR technology is used in this paper to identify and monitor the surface deformation of a large-scale mining field in the study area. The interferometric phase of the DS coherent target is usually affected by temporal decorrelation and noises. In order to suppress the influence of noises on the coherence of the DS target and improve the signal-to-noise ratio of the DS target, its interferometric phase should take the homogeneous filtering with the spatial adaptiveness to improve its quality. However, after the interference phase of the DS target is filtered, its phase coherence is destroyed, and it is necessary to further optimize the phase using Phase linking (PL) technology to obtain the optimal estimation of the DS interferometric phase.

In this study, the KS test algorithm was used to identify statistically homogeneous pixels (SHPs), which were then processed via homogeneous filtering in order to improve the signal-to-noise ratio (SNR) of the homogeneous pixels, and a higher quality of interferogram was obtained.

It is assumed that among the time sequence of SAR images, one of them is selected as the main reference image, and the rest are slave images. After resampling, its cumulative distribution function is as follows: (1)$$F\left(x\right)=\left\{\begin{array}{cc} 0 & \;x\leq d_{1}\\ \frac{k}{N} & \;d_{k}\leq x\leq d_{k+1}\\ 1 & \;x\geq d_{N} \end{array}\right.$$
where $d_{k}$ is the amplitude value of the pixel on image *k* and F(x) is the cumulative distribution function. Assume that any two pixels in the test are *m*, *n*, respectively, and the statistical magnitude of the KS test can be expressed as: (2)$$D_{KS}=\max\left|F_{{}_{m}}\left(x\right)-F_{{}_{n}}\left(x\right)\right|$$
where $F_{{}_{m}}\left(x\right)$ and $F_{{}_{n}}\left(x\right)$ are cumulative distribution functions corresponding to the pixels, respectively. At the significance level α, the rejection domain of the KS test is: (3)$$D_{\alpha }>\sqrt{-\frac{1}{N}\ln\left(\frac{\alpha }{2}\right)}$$
where the critical value under the significance level α is expressed as $D_{\alpha }$. When the statistical magnitude of *m*, *n* is $D_{KS}<D_{\alpha }$, *m*, *n* are accepted as homogeneous pixels.

In this paper, a PL algorithm, referred to as an eigendecomposition-based maximum-likelihood-estimator of Interferometric phase (EMI) algorithm [23] is used to optimize the phase of the DS coherent targets. The covariance matrix *C* corresponding to the DS pixel in *N* images of SAR is expressed as: (4)$$C=E\left[yy^{H}\right]=\frac{1}{M}\sum_{y\in \Omega }yy^{H}$$
where *E* is mathematical expectation; $y={\left[y_{1},y_{2},\cdots ,y_{n}\right]}^{T}$ is the complex reflectance of the DS pixel; Ω is the homogeneous region; *M* is the number of pixels contained in a homogeneous region; and *H* is a conjugate transpose.

In order to suppress the adverse effects of the interferogram with low coherence on phase estimation, the EMI algorithm assigns different weights to the interferogram based on their coherence quality. The optimal estimation $\hat{\theta }$ of the DS phase is expressed as follows: (5)$$\begin{array}{cc} & \hat{\theta }=\arg\min_{\theta }\left\{\mu^{H}\left(\Gamma^{-1}\circ C\right)\mu \right\}\\ & =\arg\min_{\theta }\left\{\sum_{i=1}^{N}\sum_{j>i}^{N}\gamma_{i,j}^{\prime }c_{i,j}\mu_{i,j}^{\prime }\cos\left(\varphi_{i,j}-\theta_{i,j}\right)\right\} \end{array}$$
where ∘ is the Hadamard product; Γ is the coherence matrix of the DS; $\gamma_{i,j}^{\prime }={\left[\Gamma^{-1}\right]}_{i,j}$ is the weight regulation factor of each interferogram; $C_{i,j}={\left[|C|\right]}_{i,j}$; μ is the eigenvector corresponding to the minimum eigenvalue; $\mu_{i,j}^{\prime }=[\left|\mu \right|{\left|\mu \right|}^{T}{]}_{i,j}$ is the observed value of the interferometric phase of the DS target; and $\theta_{i,j}=\theta_{i}-\theta_{j}$.

In order to evaluate the phase optimization of coherent DS targets and the influence of the decoherence effect subject to time, the fit degree $\gamma_{{}_{EMI}}$ of the phase estimation is calculated according to Equation (Equation 6) based on the initial and optimized interference phase of the coherent DS targets. For pixels whose fitting of phase is better than the set threshold value, the optimized interference phase is used to replace the initial one, and is identified as the DS point.
(6)γEMI=2N(N−1)Re{∑i=1N∑j=i+1Nexp(jφi,j)exp[−j(θ^i−θ^j)]}
where Re{} is the real part of the complex number calculation; $\varphi_{i,j}$ is the initial interference phase between the SAR image; and ${\hat{\theta }}_{i},{\hat{\theta }}_{j}$ are the optimized phases of the SAR images *i* and *j*. After the fusion of PS and DS in the study area, it is only essential to further adopt the processing of the time sequence of PS-INSAR to obtain high-precision surface deformation monitoring results, and the data processing is shown in Figure 3.

## 3. Results and Discussion

### 3.1. InSAR Rate of Deformation

The average annual subsidence rate of the study area was obtained by applying DS-InSAR technology from January 2018 to December 2020 (Figure 4). The value of the subsidence rate shown in the figure was the result of deformation monitoring along the line of sight (LOS) of the satellite. Based on the referring point of phase unwinding, the positive and negative values of the deformation rate indicate that the target of the coherent point extracted using DS-InSAR technology was moving to or away from the satellite, respectively. As can be seen from Figure 4, from January 2018 to December 2020, the surface of the four groups of mines in the study area was relatively stable as a whole, and the surface of a few small areas showed a trend of subsidence, with an annual average subsidence rate of about −19.6 mm/a∼20 mm/a. In the group of mines labeled A, there were two obvious places of funnel-shaped subsidence (Figure 4A), and the cause of surface subsidence in area A-1 is still unclear. Area A-2 includes the Tuzhu Coal Mine and Yipingdong Coal Mine, and it was preliminarily determined that the surface subsidence in this area is caused by underground coal mining. There was a trend of subsidence (C1∼C5) in many places within the group of mines labeled C, which has mining fields around the places showing subsidence. It was preliminarily concluded that the surface subsidence in these places is caused by underground coal mining. Among them, many mining fields in C-1 and C-2 are clustered, and the funnel-shaped subsidence in the C-3 mining area was the most obvious one (Figure 4C). In both the B and D groups of mines, the surface remained stable as a whole (Figure 4B,D), and no obvious trend of surface subsidence was found around their mining areas. This is probably due to the dense coverage of vegetation or the high rate of surface deformation in these mine fields, which makes it difficult to extract the coherent points, resulting in poor monitoring effects.

### 3.2. Surface Deformation in Mining Areas

#### 3.2.1. Surface Deformation in Group of Mines A

The DS-InSAR monitoring results of the surface subsidence of group of mines A were amplified and are shown in Figure 5a. The area A-1 showed a relatively obvious trend of subsidence (Figure 5b), and the maximum annual surface subsidence rate of LOS in this area was about −18.5 mm/a. However, the data on coal mining areas collected in this paper showed that there is no coal mine in this area, and the cause of surface subsidence is unclear. In order to further determine the cause of surface subsidence, the data of four historic optical images of the A-1 region were investigated with the help of Google Earth (Figure 6a–d). As can be seen in Figure 6a–c, during the period from 20 October 2014 to 29 October 2018 there was no significant change in the land surface within the solid red line of area A-1. However, from the images taken on 8 November 2020, it can be seen that the land surface inside the solid red line in area A-1 had changed significantly and showed signs of being mined. The geographical location of area A-1 belongs to Xikuang Mountain, Hunan Province, where antimony ore is abundant; therefore, it can be inferred that the phenomenon of surface subsidence in this area may be caused by the mining of metal ores after 29 October 2018.

In the data of cumulative surface subsidence with the time sequence extracted using DS-InSAR technology, nine time nodes were selected to plot the development process of the deformation field (Figure 7). The surface of area A-1 (Figure 7) remained stable as a whole. Only a small number of coherent targets showed a trend of sinking, and the funnel-shaped subsidence had not yet begun to develop on 1 May 2019. Since 4 October 2019, the surface of region A-1 had begun to show a slight trend of subsidence, and then the funnel-shaped subsidence began to gradually form. Over time, the funnel-shaped subsidence gradually expanded from the suspected mined area to the northwest and northeast. Until the end of the monitoring on 27 November 2020, the funnel-shaped surface subsidence was still in a developing state, and its scope of influence was further expanded compared to that on 16 September 2020. In addition, combined with the historic optical images of area A-1, it can be seen that there were no suspected signs of metal ore being mined in the area by the end of 29 October 2018. Instead, it should start between 29 October 2018, and 8 November 2020. Therefore, the inverted development of the funnel-shaped subsidence in A-1 based on the DS-InSAR deformation monitoring results was basically consistent with the phenomenon presented in optical images from the perspective of time, and it is clear that the phenomenon of surface subsidence in A-1 was caused by metal mining activities.

Within the group of mines referred to as A, surface subsidence also occurred in A-2. To the northeast of this area lies the Tuzhu Coal Mine, and two elliptically funnel-shaped areas of subsidence were found around the mine. To its southwest lies the Yipingdong Coal Mine, and the maximum annual surface subsidence rate of LOS was about −11.7 mm/a and −12 mm/a, respectively (Figure 5c). As can be seen from the figure, the DS-InSAR monitoring results only extracted the surface subsidence information at the edge of the mining area, while the central area presented a void state, meaning the system failed to extract sufficient deformation information. It is estimated that the decoherence effect in a void area is serious due to intense mining activities, which is in accordance with the law of surface subsidence in mining areas. However, it can be inferred from the gradient change in the subsidence rate of the edge of the two mining areas (from green to red) that the rate of the central part of the surface subsidence should be much higher than the monitored value. According to the data collected from coal mining areas in this study, the legal time for the mining of the two mining areas is from 25 September 2019 to 25 September 2024. In order to clarify whether there was illegal mining before the legal time, nine time nodes were selected from the data on the cumulative surface subsidence of the time sequence extracted using the DS-InSAR technology to plot the development process of its deformation field (Figure 8). The monitoring results of the cumulative surface subsidence of the time sequence showed that on 10 September 2019, the surrounding surface of the two mining areas remained stable as a whole, without an obvious trend of subsidence. Therefore, it was concluded that no artificial mining activities were found in the mining area, which is also consistent with the related data collected in the mining area from the angle of time. The Yipingdong Coal Mine, located in the southwest, began to show a slight trend of subsidence on 25 February 2020, while the Tuzhu Coal Mine, in the northeast, showed a slight trend of subsidence as of 20 March 2020. Since then, the funnel-shaped subsidence started to develop gradually. With continuous mining, the boundary of funnel-shaped subsidence gradually became evident, and the scope of influence also gradually expanded. By 28 September 2020, the funnel-shaped surface subsidence had developed further. Therefore, it can be determined that no mining activities occurred before then according to the reverted development process of funnel-shaped subsidence found from the data of DS-InSAR monitoring in the two mining areas.

#### 3.2.2. Surface Deformation in Group of Mines C

The DS-InSAR monitoring results of the surface subsidence of the group of mines referred to as C were amplified (Figure 9). In most areas of the group, the surface remained relatively stable as a whole, and the annual average surface subsidence rate of LOS was about −19.6 mm/a∼20 mm/a. A trend of surface subsidence is shown in five parts of the area (Figure 9a–e), where for the smaller parts, the specific information with the subsidence is shown in Table 2, including the corresponding period of legal mining, the annual surface subsidence rate of LOS, and the magnitude of cumulative subsidence. Among the five parts of the area (Figure 9a–e), the Zhouyuanshan Coal Mine in area D presented the most obvious funnel-shaped subsidence, with railways crossing the affected area. Therefore, further analysis will be conducted to focus on the surface subsidence along the railway in the mining area.

The maximum surface subsidence rates of LOS in the No. 1 Daozi Mine and the Leiyang and Yuanjiang Mountain Coal Mine distributed in area A were −17 mm/a and −10 mm/a (Figure 9), respectively, and they were converted into cumulative deformation variables of about −51 mm and −30 mm, respectively, according to the span of the monitoring time. However, it is worth noting that the mining rights of No. 2 Daozi Mine and Yuanjiang Mountain Coal Mine came into effect on 21 September and 28 August 2019, respectively, and the span of the monitoring time only covered 2 to 3 months of their authorized mining time. Considering that the above monitoring value is only a reflection of the surface subsidence on the edge of the mining area, the magnitude of subsidence in the real mining area should be much larger than the monitoring value. Due to the large surface subsidence in the non-central mining area in a short period of time, there may have been early mining or illegal mining in the two mining areas. Based on the news released publicly on the network by Hunan Branch, the China Mine Safety Administration showed that a major flooding accident occurred in the Yuanjiang Mountain Coal Mine, Daozi Coal Company, Leiyang, on 29 November 2020, resulting in 13 deaths and an economic loss of about CNY 34.84 million. The official report of the accident showed that the mining of the Yuanjiang Mountain Coal Mine and No. 2 Daozi Mine was excessively deep and beyond the allowed boundary, which led to this major safety-related accident. Therefore, the surface deformation monitoring results extracted based on DS-InSAR data are of reference value in the inspection of illegal mining and prevention of safety-related accidents.

According to the relevant data collected in the mining area, the Zhouyuan Mountain Coal Mine is located in Zixing City, Hunan Province, with a reserve of coal of 18.1 million tons. The validity period of its mining rights was from 30 March 2016 to 30 March 2021; a railway named Xusan runs through the mining area, a section about 2.4 km long. The monitoring results of the surface subsidence in the Zhouyuan Mountain mining area and the boundary of the mining area are shown in Figure 10. In the northwest of the mining area, large funnel-shaped subsidence appeared, and the maximum subsidence rate of the surface of the LOS was about −18.9 mm/a. From the coherent target extracted from the north of the mining area, it can be seen that the surface subsidence presented a gradient change, and starting from the edge of the mining area, the magnitude of the surface subsidence increased gradually as it neared the mining area. The subsidence monitoring results were consistent with the subsidence law of mining areas.

A field investigation was carried out in the mining area with funnel-shaped subsidence to further clarify whether underground coal mining in the Zhouyuan Mountain mining area had caused harm to the life and property of the residents there. The investigation revealed that some houses, buildings, and roads scattered within the funnel-shaped subsidence of the Zhouyuan mining area had cracked (Figure 11). These cracks were attributed to the loss of ground material and the massive extraction of groundwater in the process of underground coal mining, which led to the weakening of the bearing capacity of the ground, and then to surface subsidence due to the gradient change in the surface subsidence in the Zhouyuan Mountain mining area. In addition, the mining degree and groundwater level in the mining area were often not consistent, so there were differences in the bearing capacity of the strata, which caused uneven surface subsidence of the roads and houses, resulting in cracking or structural damage. The cracked houses shown in Figure 11a–d and the damaged roads shown in Figure 11e,f indicate that the mining in the Zhouyuan Mountain mining area has threatened the safety of some houses and roads.

Considering that the Xusan Railway passes through the funnel-shaped subsidence monitored using DS-InSAR technology, the coherent targets were screened within the 100 m buffer zone along the railway, and the surface subsidence along the railway was analyzed (Figure 12). Based on the surface subsidence along the railway, it can be seen that the north of the railway did not pass through the mining area, and the surface remained stable as a whole and was not affected by mining. Sections D1–D5 along the railway passed through the Zhouyuan Mountain mining area successively. As the railway approaches the mining area, the land surface subsidence shows a trend of accelerating. To the end of section D1, the Xusan Railway passes through the mining area, a section about 650 m long, and there, the maximum surface subsidence rate of LOS was about −7.4 mm/a. At the end of Section D2, the Xusan Railway passes through the mining area, a section about 1180 m long, and the subsidence rate accelerated with the increase in the crossing range, about −9.9 mm/a. Towards the end of section D3, the railway further penetrates the mining area, 1600 m, and the subsidence rate further accelerated to −16.2 mm/a. At the end of section D4, the railway passes through the mining area, about 2020 m, and the surface subsidence rate along the railway reached its peak value, −18.9 mm/a. The railway in D5–D6 gradually approaches and crosses the edge of the mining area, and the surface subsidence rate slowed down here, about −13.8 mm/a.

In order to reflect the changes in the land subsidence rate more straightforwardly along the Xusan Railway, samples were collected about every 100 m from the northern end of the buffer zone along the railway, and the change curve of the subsidence rate along the buffer zone was drawn (Figure 13). From the changing trend along the railway, we determined that the magnitude of the surface subsidence rate of the 0–1000 m section was very small, which may be caused by the periodic load imposed by the train during its operation. In the 1000–3400 m section along the Xusan Railway, the land subsidence rate continued to accelerate, reaching a peak of −18.2 mm/a at 3400 m, and then it gradually slowed down.

In order to further evaluate the influence of subsidence caused by the railway crossing the mining area, 19 monitoring points (P0–P18) were selected to analyze the magnitude of the cumulative deformation of the time sequence at the railway section with severe subsidence based on the sampling at intervals of 100 m. The monitoring points are shown in Figure 14 and the dynamic curve of the cumulative subsidence at the monitoring points is shown in Figure 14. Based on the dynamic subsidence curve, it can be seen that all monitoring points showed a trend of subsidence, among which the trend of adjacent subsidence of P0–P10 monitoring points was more well distributed, and the gradient differences in the subsidence of the adjacent monitoring points was not large. However, the differences in the adjacent subsidence of the P11–P18 monitoring points was evident, with a large gradient subsidence, which may affect the safe operation of the railway.

In order to reflect the dynamic development process of funnel-shaped subsidence more straightforwardly and also reflect the spatial distribution and influenced range of the surface subsidence in different periods in the Zhouyuan Mountain mining area, nine time nodes were selected during the monitoring period to show the development process of the deformation field (Figure 15). The monitoring results of the cumulative surface subsidence of the time sequence showed that a small part of the mining area presented an obvious upward trend on 31 March 2018 due to the fact that the trend of subsidence of the deformation field of the time sequence was determined using the main images on 1 January 2019. Compared with both the time points on 31 March 2018 and 27 September 2018, the surface of the Zhouyuan Mountain mining area was already in a subsiding trend on 1 January 2019. Therefore, it is reasonable for the surface of the mining area to show an upward trend at those two particular time points in 2018 (Figure 15). With the continuous mining activities, the gradient change in surface subsidence began to show on the northern edge of the mining area on 25 December 2019. After that, the funnel-shaped surface subsidence continued to develop, the influenced range of surface subsidence also continued to expand, and the gradient classification of the surface subsidence deformation became more and more evident.

## 4. Discussion

Due to the differences in the specific mining times of each mining area, when the monitoring period covered a small proportion of the mining time, some coal mining areas showed a small magnitude of surface subsidence. For example, in the Tuzhu Coal Mine and Yipingdong Coal Mine in A-2, the maximum annual surface subsidence rate of LOS in the two mining areas was no more than −15 mm/a. In addition, the dense vegetation in the study area, coupled with changes in the underground structure of some mining areas where the open-cast mining occurred, led to difficulties in the extraction of coherent points using DS-InSAR technology, which may result in omissions in the identification of groups A and C, and may also be the main reason for the failure to identify the surface subsidence of the mining areas in groups of mines B and D. Finally, due to the lack of field test data in the mining area to verify DS-InSAR results, and the lack of a Sentinel-1A descent orbit image dataset in the study area, the verification could not be achieved using results of different orbits. However, the period during which the surface subsidence occurred in the mining area coincided well with the mining period. Moreover, the field investigation in the Zhouyuan Mountain mining area showed that the buildings and roads in the funnel-shaped subsidence area had been destroyed, which confirms the reliability of the DS-InSAR monitoring results obtained in this study.

## 5. Conclusions

DS-InSAR technology was employed to identify and monitor the surface subsidence of four groups of coal mines in Hunan Province, China, where the density of vegetation coverage is high and most coal mine areas are distributed at the edges of villages or mountainous areas. In the whole monitoring process, the data of 86 Sentinel-1A SAR images from January 2018 to December 2020 were used to extract the annual average rate of surface subsidence of LOS, with the rate ranging from −19.6 mm/a to 20 mm/a. The results show that surface subsidence was not only detected in two coal mining areas in group A, but also in 1 metal mining area; this was found by combining their data with data from optical images, and we also detected 11 coal mining areas in group C, in 2 of which safety-related accidents have occurred. In addition, the development process of surface subsidence was also inverted in four typical mining areas, and the influence of surface subsidence on the Xusan Railway crossing area of the Zhouyuan Mountain Coal Mine was evaluated. The development time of the surface subsidence in three typical coal mining areas was consistent with the legal time limit for mining, and in one metal mining area it was also consistent with the signs of mining activities seen in the optical images. Moreover, the field investigation results of the Zhouyuan Mountain coal mining area also showed that houses and roads in the mining area have been damaged by the impact of surface subsidence, which verifies the reliability and validity of the results in this paper. As a result, the application of DS-InSAR technology in monitoring mining areas can effectively realize the identification and monitoring of their surface subsidence on a large scale, and the results can help the local government build a map of surface deformation in mining areas in Hunan and provide a reference for resource management, the prevention of safety-related accidents, combating illegal mining activities for mineral resources, etc.

## Figures and Tables

**Figure 1 sensors-23-08146-f001:**
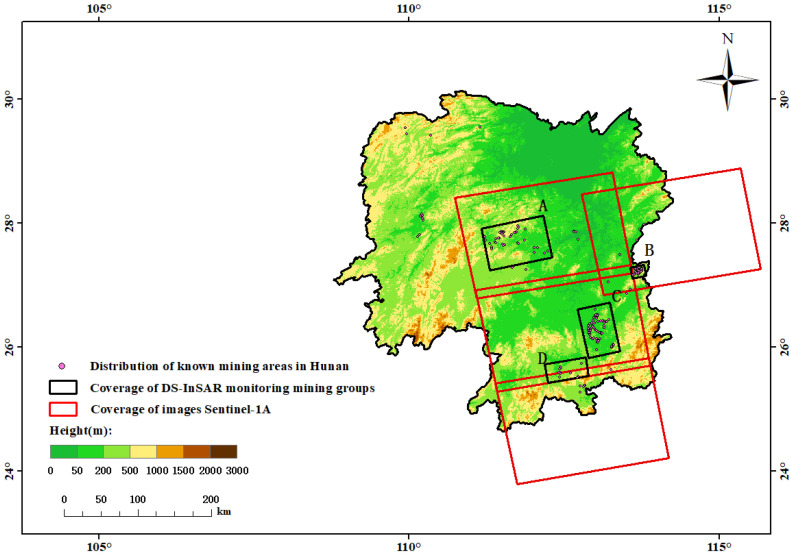
Sentinel-1A coverage of image data and distribution of mining fields.

**Figure 2 sensors-23-08146-f002:**
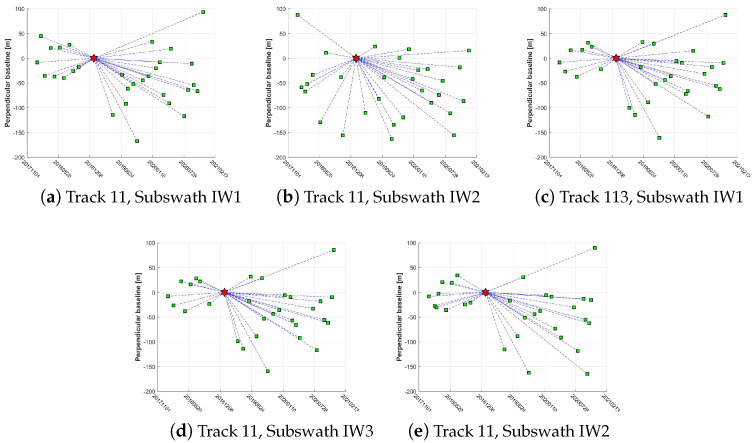
Spatiotemporal baseline information of the interference pairs (the red star and green square denote the master and slave SAR image, respectively).

**Figure 3 sensors-23-08146-f003:**
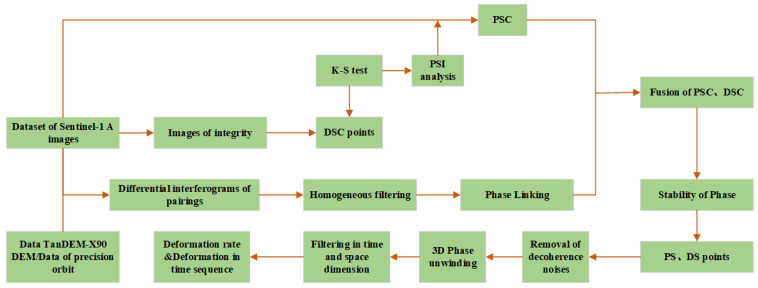
DS-InSAR data processing.

**Figure 4 sensors-23-08146-f004:**
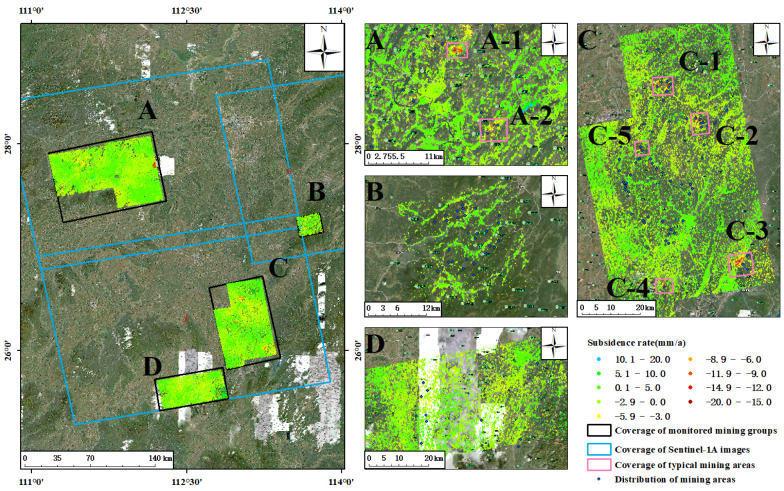
Surface subsidence monitoring results of DS-InSAR in A, B, C, and D groups of mines in Hunan Province. (**A**–**D**) are the deformation rates of A–D mining areas, respectively.

**Figure 5 sensors-23-08146-f005:**
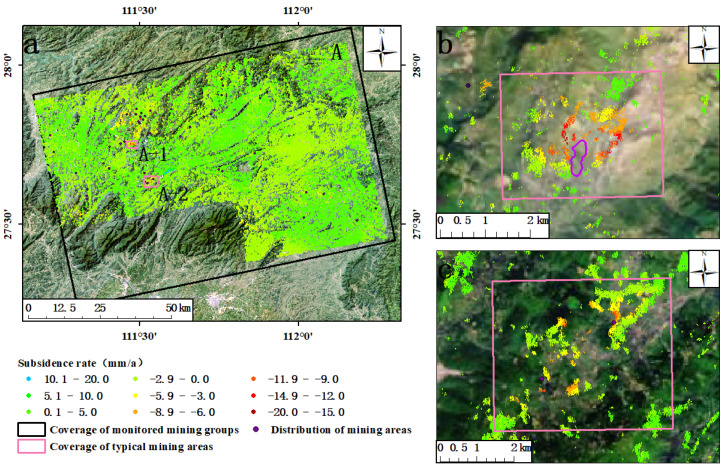
A: Deformation rate of mining area A. (**a**) The overall deformation rate of mining area A. (**b**) The deformation rate of Xikuang Mountain antimony ore. (**c**) The deformation rate of Tuzhu Coal Mine and Yipingdong Coal Mine.

**Figure 6 sensors-23-08146-f006:**
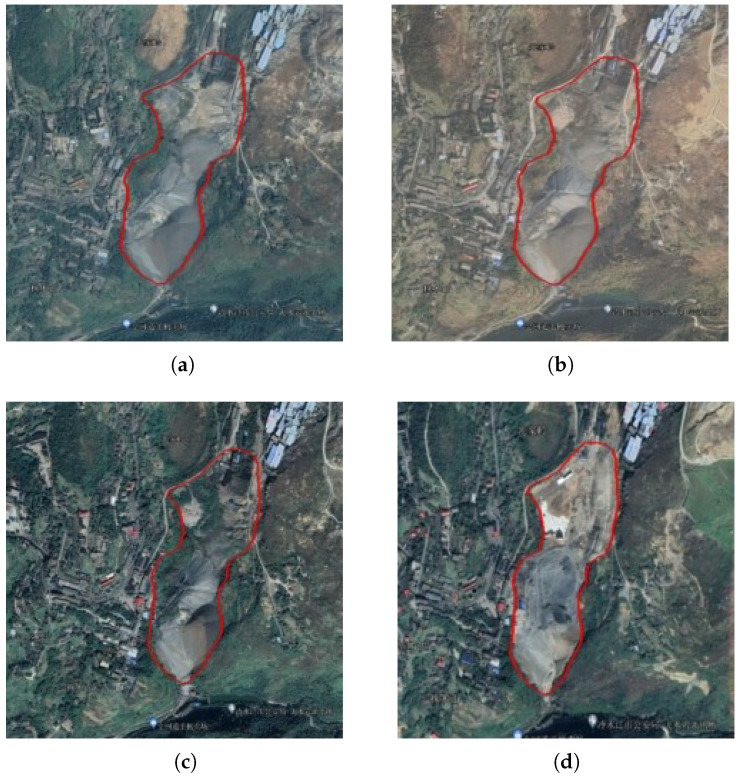
A-1: Historic optical images of area A-1. (**a**) An optical image of area A-1 taken on 20 October 2014. (**b**) An optical image of area A-1 taken on 23 January 2017. (**c**) An optical image of Area A-1 taken on 29 October 2018. (**d**) An optical image of area A-1 taken on 8 November 2020.

**Figure 7 sensors-23-08146-f007:**
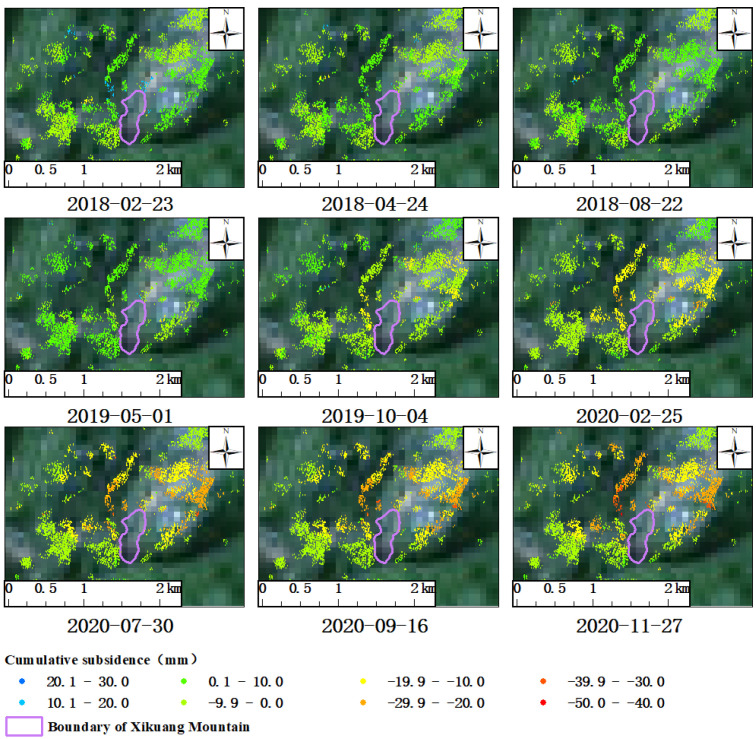
A-1 (Xikuang Mountain antimony ore): cumulative deformation field of dynamic time sequence.

**Figure 8 sensors-23-08146-f008:**
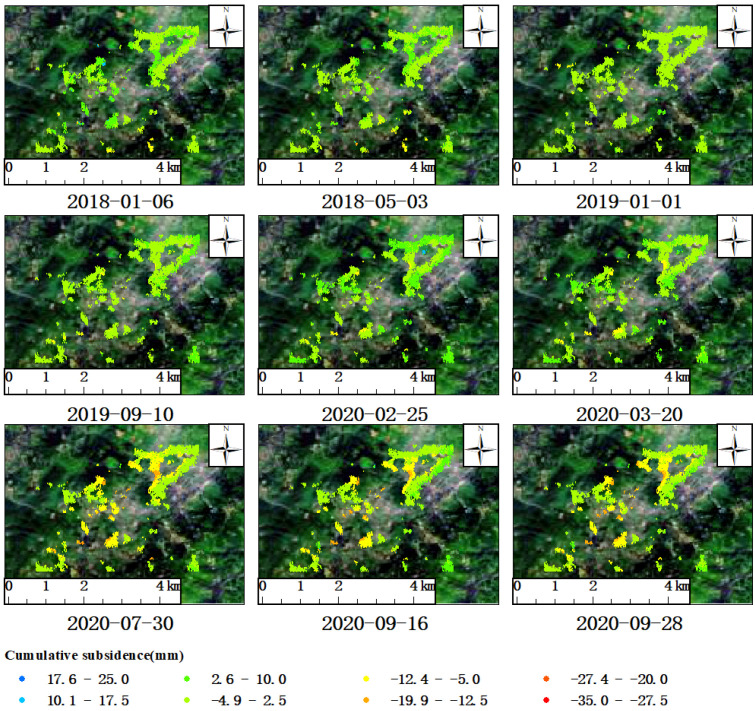
A-2 (Tuzhu and Yipingdong Coal Mine): cumulative deformation field of a dynamic time sequence.

**Figure 9 sensors-23-08146-f009:**
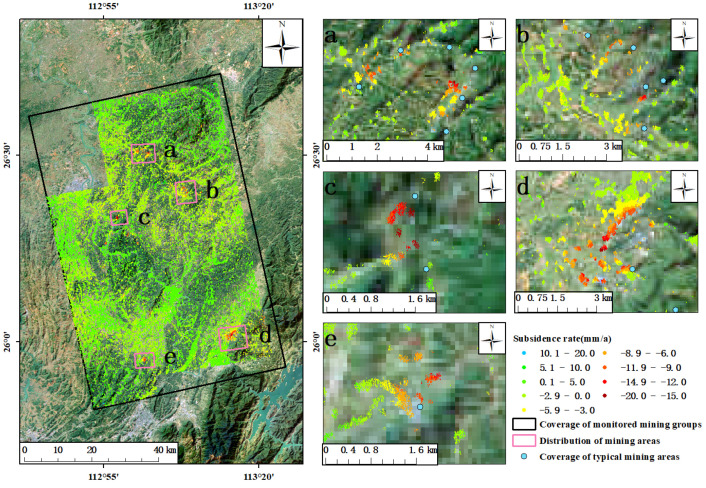
Deformation rate of mining area C. Smaller parts (**a**–**e**) of area C represent the rates of deformation.

**Figure 10 sensors-23-08146-f010:**
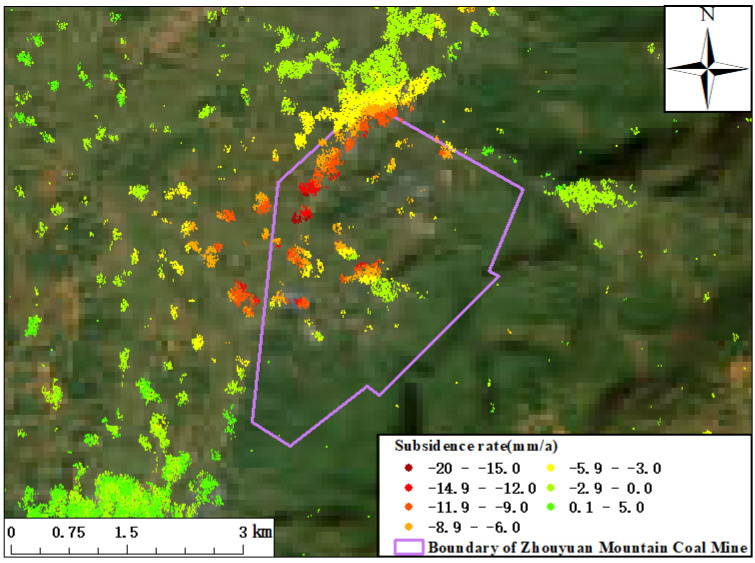
Surface deformation rate of Zhouyuan Mountain Coal Mine.

**Figure 11 sensors-23-08146-f011:**
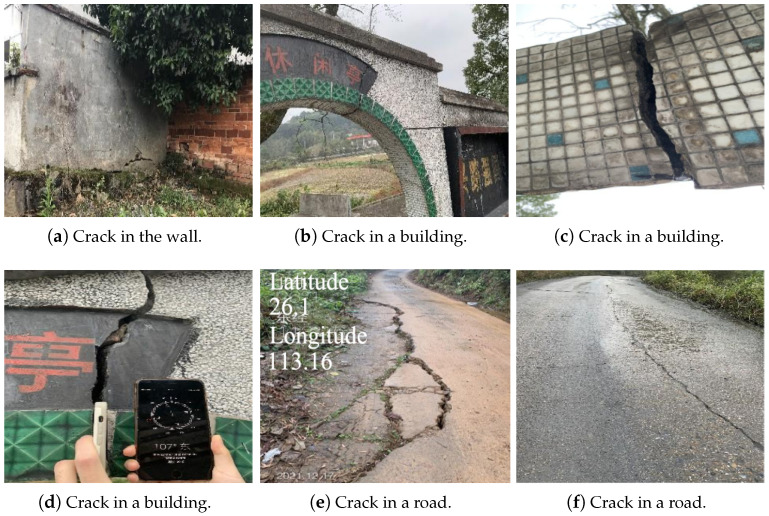
Field investigation of Zhouyuan Mountain Coal Mine. (**a**–**d**) show the damaged buildings. (**e**,**f**) show cracks in and damage to roads.

**Figure 12 sensors-23-08146-f012:**
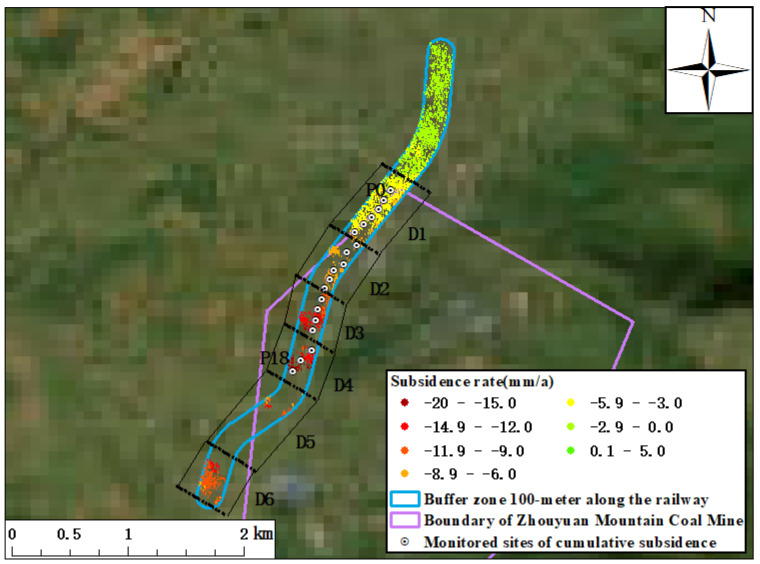
Surface deformation rate in the 100 m buffer zone along Xusan Railway.

**Figure 13 sensors-23-08146-f013:**
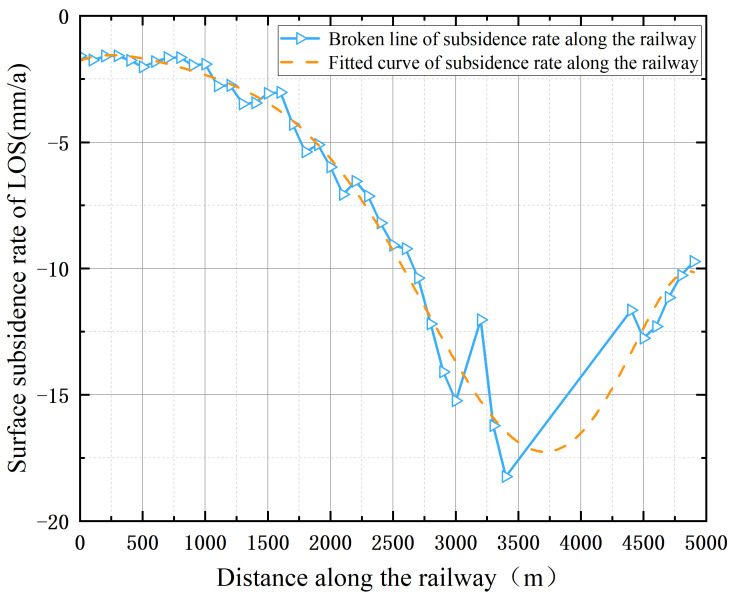
Curve of surface deformation rate along Xusan Railway.

**Figure 14 sensors-23-08146-f014:**
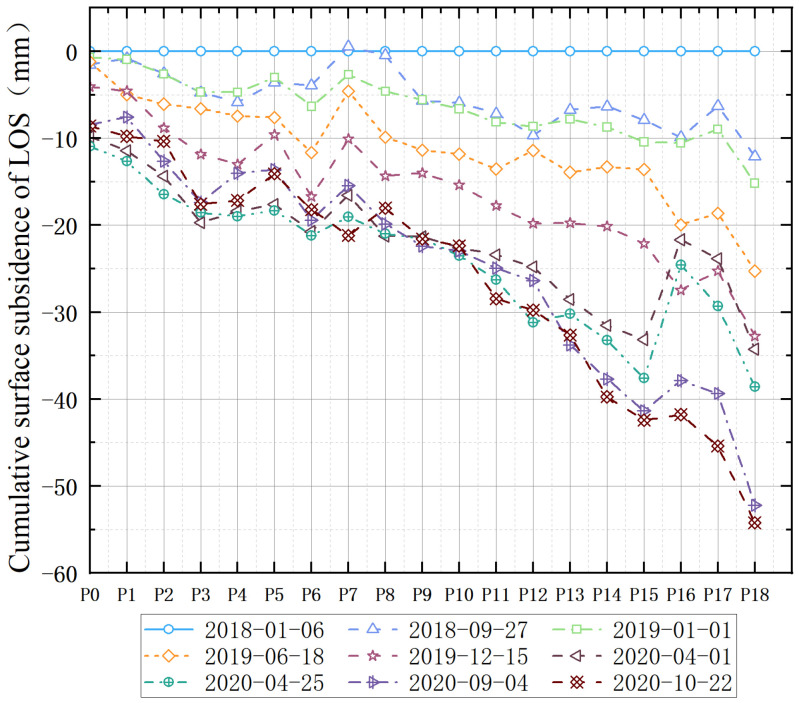
Dynamic broken line of cumulative surface deformation magnitude along Xusan Railway.

**Figure 15 sensors-23-08146-f015:**
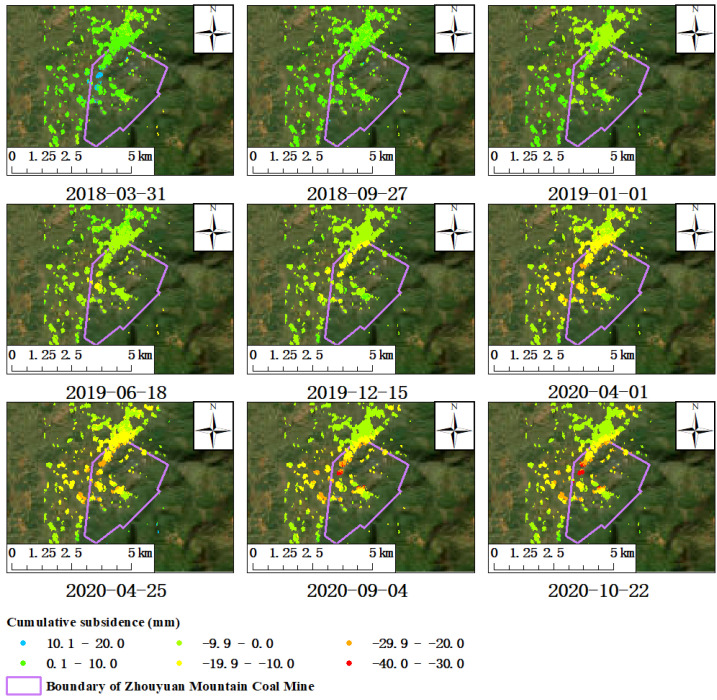
Cumulative deformation field of dynamic time sequence in Zhouyuan Mountain mining area.

**Table 1 sensors-23-08146-t001:** Metadata about the Sentinel-1 data used in the study.

Parameters	Area			
	**A**	**B**	**C**	**D**
Dataset	Sentinel-1A	Sentinel-1A	Sentinel-1A	Sentinel-1A
Track	11	113	11	11
Subswath	IW1, IW2	IW1	IW3	IW2
Numer	31	31	34	31
Orbit mode	Ascending	Ascending	Ascending	Ascending
Heading (degree)	347	347	347	347

**Table 2 sensors-23-08146-t002:** Mining locations and schedule.

Names of Mine Coal	Aera	Duration of Mining	Subsidence Rate	Cumulative Subsidence
Motian	a	2019-11-05∼2024-11-19	−13 mm/a	−39 mm
Fengxing	a	2019-09-24∼2020-09-24	−11 mm/a	−33 mm
No. 2 Daozi	a	2020-09-21∼2021-09-21	−17 mm/a	−51 mm
Yuanjiang Mountain	a	2020-08-28∼2021-08-28	−10 mm/a	−30 mm
Mingchong Coal Co.	b	2019-12-14∼2020-12-14	−12 mm/a	−36 mm
Tanshan	b	unknown	−15 mm/a	−45 mm
Dongyuan	b	unknown	−9 mm/a	−27 mm
Feijiang	c	2018-08-31∼2023-08-31	−19.6 mm/a	−58.8 mm
Hengfa	c	2019-07-24∼2020-07-24	−16.8 mm/a	−50.4 mm
Zhouyuan Mountain	d	2016-03-30∼2021-03-30	−18.9 mm/a	−56.7 mm
Chashanling	e	2019-02-11∼2023-02-11	−15 mm/a	−45 mm

## Data Availability

Not applicable.

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
