# Peer review of "Surface Subsidence Monitoring of Mining Areas in Hunan Province Based on Sentinel-1A and DS-InSAR"

_sensors, 2023, doi:10.3390/s23198146_

Round 1

Reviewer 1 Report

In this paper, DInSAR was used to monitor surface settlement in mining areas in Hunan Province, and the technical method adopted was feasible. DSInSAR is suitable for the study of land subsidence in forested areas, but the paper still has the following problems and needs to be further improved.

(1) A map of the location of the study area should be given in the introduction section;

(2) The abbreviation of some papers does not give the full name, and is included in InSAR, GNSS.

(3) The meaning of some variables is not clear, such as α  γ_EMI;

(4) It is important to compare the InSAR results with in-situ measurement, e.g, time-series GNSS data or level results;

(5) PSs/DSs are in A~D area number should be given in Sec. 3.
(6)
Discussion part is important to grasp the key results of the research and analyze it in depth, explain and sublimate the originally meaningless data, facts, etc., so as to give answers to research questions. It is necessary to further elaborate on the advanced nature of research methods and discuss the significance of research

Author Response

Dear Reviewer,

We gratefully thank you for your affirmation to our work. Based on the instructions provided in your letter, we uploaded the file of the revised manuscript. Appended to this letter is our point-by-point response to the comments, followed by a list of all relevant changes made in the manuscript and the marked-up manuscript version.

In this paper, DInSAR was used to monitor surface settlement in mining areas in Hunan Province, and the technical method adopted was feasible. DSInSAR is suitable for the study of land subsidence in forested areas, but the paper still has the following problems and needs to be further improved.

Comments 1: A map of the location of the study area should be given in the introduction section;

Response 1: Thanks a lot. The map of the location of the study area has given in the Sect. 2.1 (Figure 1 in page 3).

Comments 2: The abbreviation of some papers does not give the full name, and is included in InSAR, GNSS.

Response 2: Thanks a lot. We have supplemented the full name of some abbreviations as following:

InSAR: interferometric synthetic aperture radar (page 1, paragraph 1, line 6)

GNSS: global navigation satellite system (page 2, paragraph 1, line 2)

PL: phase linking (page 5, paragraph 5, line 7)

EMI: eigendecomposition-based maximum-likelihood-estimator of interferometric phase (page 5, paragraph 5, line 8)

Comments 3: The meaning of some variables is not clear, such as α  γ_EMI;

Response 3: Thanks a lot. The α denotes the significance level, and the γEMI is a measure of “goodness of fit” between the estimate and original data. Besides, we have supplemented the meaning of variables in each equation.

Comments 4: It is important to compare the InSAR results with in-situ measurement, e.g, time-series GNSS data or level results;

Response 4: Thanks a lot. It’s exactly as you said that in-situ measurements (collected from GNSS or precise leveling) are generally used to evaluate the accuracy and precise of InSAR results.

Unfortunately, no in-situ data is publicly available in the study area until now.

Therefore, we assessed the consistency of InSAR results by a cross comparison between different datasets, such as optical images and filed survey, which suggests the availability and reliability of InSAR results.

Comments 5: PSs/DSs are in A~D area number should be given in Sec. 3.

Response 5: There are 3457746, 87986, 1462325 and 945066 of PSs and DSs in area A, B, C and D, respectively.

 We have supplemented the number in Sect. 3.1 (page 6, paragraph 3, line 8).

Comments 6: Discussion part is important to grasp the key results of the research and analyze it in depth, explain and sublimate the originally meaningless data, facts, etc., so as to give answers to research questions. It is necessary to further elaborate on the advanced nature of research methods and discuss the significance of research

Response 6: Thanks a lot. The significance of the research is elaborated in detail in Sect. 3. Therefore, the Sect. 4 has been merged into the Sect. 3 which has changed into “3. Results and discussion”.

Reviewer 2 Report

1. The title can be modified as “Mining Subsidence Monitoring in Hunan Province Based on DS-InSAR”.

2. Please provide a description (e.g, rationale and scope) of DS techniques adopted by this text.

3. Please describe the dataset used with a table resuming all the main technical features of Sentinel dataset in Sec. 2.2.

4. Please provide the values adopted for the α,γEMI, the size of rectangular window, and the threshold used to identify SHP. How to chose these values?

5. The title of Figure 3 may be changed as “DS-InSAR processing flow chart”.

6. The title of Figure 8 can be modified as “A-2(Tuzhu and Yipingdong Coal Mine) time series land subsidence”

7. The title of Figure14 can be modified as “… lines …..”.

8. The authors can add some research references of China University of Mining and Technology.

Minor editing of English language required

Author Response

Dear Reviewer,

We gratefully thank you for your affirmation to our work. Based on the instructions provided in your letter, we uploaded the file of the revised manuscript. Appended to this letter is our point-by-point response to the comments, followed by a list of all relevant changes made in the manuscript and the marked-up manuscript version.

  1. The title can be modified as “Mining Subsidence Monitoring in Hunan Province Based on DS-InSAR”.

Response 1: Thanks a lot. The suggestion is accepted and the title has been modified as “Mining Subsidence Monitoring in Hunan Province Based on PS/DS-InSAR”.

  1. Please provide a description (e.g, rationale and scope) of DS techniques adopted by this text.

Response 2: Thanks a lot. We have supplemented a brief description of DS in Sect. 2.3 (page 4, paragraph 2, line 7-15): “DS is usually corresponding to bare soils and widespreaded in natural landscapes. However, the interferometric phase of the DS is usually affected by temporal decorrelation. In order to improve the signal-to-noise ratio, the interferometric phase of DS should take the homogeneous filtering with the spatial adaptiveness. After the interference phase of the DS target is filtered, its phase coherence is destroyed, and it is necessary to further optimize the phase using phase linking (PL) technology to obtain the optimal estimation of the DS interferometric phase.”

  1. Please describe the dataset used with a table resuming all the main technical features of Sentinel dataset in Sec. 2.2.

Response 3: Thanks a lot. We have supplemented the brief information of the datasets in Table 1 (page 4).

Table 1. Metadata about the Sentinel-1 data used in the study.

Parameters

Area

A

B

C

D

Dataset

Sentinel-1A

Sentinel-1A

Sentinel-1A

Sentinel-1A

Track

11

113

11

11

Subswath

IW1, IW2

IW1

IW3

IW2

Numer

31

31

34

31

Orbit mode

Ascending

Ascending

Ascending

Ascending

Heading (degree)

347

347

347

347

  1. Please provide the values adopted for the α,γEMI, the size of rectangular window, and the threshold used to identify SHP. How to chose these values?

Response 4: The α and γEMI is set to 0.05 and 0.6 respectively, according to the literature [23]. The SHP is identified in a rectangular window with a size of 11 by 31 (azimuth by range) and a threshold of 30 pixels.

  1. The title of Figure 3 may be changed as “DS-InSAR processing flow chart”.

Response 5: Thanks a lot. The suggestion is accepted and the title of Figure 3 has been changed as “DS-InSAR processing flow chart”.

  1. The title of Figure 8 can be modified as “A-2(Tuzhu and Yipingdong Coal Mine) time series land subsidence”

Response 6: Thanks a lot. The suggestion is accepted and the title of Figure 8 has been changed as “A-2(Tuzhu and Yipingdong Coal Mine) time series land subsidence”.

  1. The title of Figure14 can be modified as “… lines …..”.

Response 7: Thanks a lot. The suggestion is accepted and the title of Figure 14 has been changed as “Dynamic lines of cumulative subsidence along Xusan Railway”.

  1. The authors can add some research references of China University of Mining and Technology.

Response 8: Thanks a lot. We have supplemented some literatures, such as [16], published by China University of Mining & Technology to apply the InSAR to monitor the mining-induced subsidence.

Reviewer 3 Report

Dear authors, thank you for the manuscript. It regards monitoring of a mining area using InSAR method, processing 3 years of Sentinel-1 data (one track).

In addition to language, I have only minor comments:

- please use GNSS instead of GPS

- introduction, paragraph starting with "currently, many scholars" needs to be restructured and rewritten in order to contain senseful sentences

- efficiency of these methods as low -> IS low

- section 2.3 method is not comprehensible. formula (1): please explain all parameters, such as z, d1, k, G, d_k.

- section 2.3: you speak about atmospheric and decorrelation noise, and that you do "homogeneous filtering" to mitigate the noise. If I understand that "homogeneous filtering" is a kind of averaging, I agree - but this way, you do mitigate only the temporal decorrelation, not the atmospheric noise

- section 2.3: please explain PL

- section 3.1: was close or far from the satellite -> was moving to or away from the satellite

- chapter 3 many times: amplified

Dear authors, the language has to be corrected, as well as terminology (inferrence, descent etc.) However, the main problem is the word order and prepositions, so the manuscript is mostly comprehensible.

In addition, many words are divided by hyphen -, with no reason.

Author Response

Dear Reviewer,

We gratefully thank you for your affirmation to our work. Based on the instructions provided in your letter, we uploaded the file of the revised manuscript. Appended to this letter is our point-by-point response to the comments, followed by a list of all relevant changes made in the manuscript and the marked-up manuscript version.

Dear authors, thank you for the manuscript. It regards monitoring of a mining area using InSAR method, processing 3 years of Sentinel-1 data (one track).

In addition to language, I have only minor comments:

Comments 1: please use GNSS instead of GPS

Response 1: Thanks a lot. The GPS has been replaced by GNSS.

Comments 2: introduction, paragraph starting with "currently, many scholars" needs to be restructured and rewritten in order to contain senseful sentences

Response 2: Thanks a lot. The manuscript has been revised carefully according to the comments from editor and reviewers. And the language has been further polished.

Comments 3: efficiency of these methods as low -> IS low

Response 3: Thanks a lot. The spelling error has been revised.

Comments 4: section 2.3 method is not comprehensible. formula (1): please explain all parameters, such as z, d1, k, G, d_k.

Response 4: Thanks a lot. We have supplemented the meaning of these parameters (page 5, paragraph 2, line 1).

Comments 5: section 2.3: you speak about atmospheric and decorrelation noise, and that you do "homogeneous filtering" to mitigate the noise. If I understand that "homogeneous filtering" is a kind of averaging, I agree - but this way, you do mitigate only the temporal decorrelation, not the atmospheric noise

Response 5: Yes, you are right. We are sorry for this mistake. The homogeneous filter is a kind of averaging filter and can only mitigate the temporal decorrelation. The atmospheric noise is separated using a combination of temporal and spatial filter in this paper based on the assumption that atmospheric noise is uncorrelated in time but correlated in space domain.

Comments 6: section 2.3: please explain PL

Response 6: Thanks a lot. The PL is the abbreviation of phase linking which is a technique to estimate the DS phase based on a statistically homogenous pixels (SHP).

Comments 7: section 3.1: was close or far from the satellite -> was moving to or away from the satellite

Response 6: Thanks a lot. We have revised the paper based on this suggestion (page 6, paragraph 3, line 6).

Comments 8: chapter 3 many times: amplified

Response 6: Thanks a lot.

The sentence “The DS-InSAR monitoring results of the surface subsidence of group of mines A were amplified and are shown in Figure 5 (a).” has been replaced by “The deformation in area A is shown in detail in Figure 5.” (page 7, paragraph 1, line 1-2).

The sentence “The DS-InSAR monitoring results of the surface subsidence of group of mines C were amplified (Fig. a).” has been replaced by “Fig.9 shows the deformation in area C in detail.” (page 10, paragraph 1, line 1-2).

Reviewer 4 Report

The paper is about monitoring surface subsidence in mining areas in South China, by implementing DS-InSAR technique on Sentinel-1 images. It highlights the efficiency of DS-InSAR in studying surface deformation due to mining activities. The authors conducted very detailed research on surface deformation phenomena in the broader region around the mines and the railway passing through a mining area and achieved a very good correlation between mining activities and surface deformation phenomena. I need some clarifications in some points in the paper. I also have some recommendations that I hope the authors will take into consideration.

First some editing issues I identified in the paper. Many words in the paper contain a dash symbol. Could you please remove the symbol from these words if its position in the word is unnecessary. E.g. in the Introduction  envi-ronment”.

In the introduction section in the second paragraph where the methods used to monitor subsidence in the mining areas are mentioned “At present, surface subsidence monitoring and geological hazard identification in mining areas mainly rely on manual field survey, electronic total station survey, leveling and GNSS technology., if there exists literature, please cite some relevant papers.

Also, in the introduction section where you refer to InSAR studies for monitoring surface displacements in China, you could add studies about surface deformation specifically in mines in China. I send some examples

10.3390/ijerph17041170

https://doi.org/10.3390/rs15112755

In the study region section you mention that extreme precipitation events take place. Did you investigate if there is a correlation between surface deformation and the periods with extreme rainfall?

In section 2.2 Data, could you please replace ascent with ascending?

In the second line under Figure 2, replace iterferometric with interferometric.

In the sixth line under Figure 6 replace monitering with monitoring.

In Figure 8, could you please explain what the dates represent?

In table 1 the codes refer to sites in Figure 9. The subsidence column corresponds to a specific scatterer? Also, the cumulative deformation column is calculated for a specific scatterer, for the total years of InSAR analysis or for a number of scatterers near the mining area?

In figure 14, could you please explain what the lines and dates represent?

Author Response

Dear Reviewer,

We gratefully thank you for your affirmation to our work. Based on the instructions provided in your letter, we uploaded the file of the revised manuscript. Appended to this letter is our point-by-point response to the comments, followed by a list of all relevant changes made in the manuscript and the marked-up manuscript version.

The paper is about monitoring surface subsidence in mining areas in South China, by implementing DS-InSAR technique on Sentinel-1 images. It highlights the efficiency of DS-InSAR in studying surface deformation due to mining activities. The authors conducted very detailed research on surface deformation phenomena in the broader region around the mines and the railway passing through a mining area and achieved a very good correlation between mining activities and surface deformation phenomena. I need some clarifications in some points in the paper. I also have some recommendations that I hope the authors will take into consideration.

Comments 1: First some editing issues I identified in the paper. Many words in the paper contain a dash symbol. Could you please remove the symbol from these words if its position in the word is unnecessary. E.g. in the Introduction “envi-ronment”.

Response 1: Thanks a lot. We are sorry for this mistake and have carefully revised the manuscript. And the language has been further polished.

Comments 2: In the introduction section in the second paragraph where the methods used to monitor subsidence in the mining areas are mentioned “At present, surface subsidence monitoring and geological hazard identification in mining areas mainly rely on manual field survey, electronic total station survey, leveling and GNSS technology.”, if there exists literature, please cite some relevant papers.

Also, in the introduction section where you refer to InSAR studies for monitoring surface displacements in China, you could add studies about surface deformation specifically in mines in China. I send some examples

10.3390/ijerph17041170

https://doi.org/10.3390/rs15112755

Response 2: Thanks a lot. We have supplemented three literatures, e.g, [14-16], about the applications of InSAR to monitor the mining-induced subsidence (page 2, paragraph 2, line 11).

Comments 3: In the study region section you mention that extreme precipitation events take place. Did you investigate if there is a correlation between surface deformation and the periods with extreme rainfall?

Response 3: Thanks a lot. It’s interesting to investigate the correlation between the surface deformation and the extreme rainfall. Unfortunately, we only know that the precipitation of the study area is concentrated in the summer, but the accurate period of extreme precipitation is unknown.

Comments 4: In section 2.2 Data, could you please replace ascent with ascending?

Response 4: Thanks a lot. The word “ascent” has been replaced by “ascending”.

Comments 5: In the second line under Figure 2, replace iterferometric with interferometric.

Response 5: Thanks a lot. The word “iterferometric” has been replaced by “interferometric”.

Comments 6: In the sixth line under Figure 6 replace monitering with monitoring.

Response 6: Thanks a lot. We are sorry for this mistake. The word “monitering” has been replaced by “monitoring”.

Comments 7: In Figure 8, could you please explain what the dates represent?

Response 7: The dates represent the time of SAR images, e.g, the 2018-01-06 represent a SAR image acquired in January 5, 2018.

Comments 8: In table 1 the codes refer to sites in Figure 9. The subsidence column corresponds to a specific scatterer? Also, the cumulative deformation column is calculated for a specific scatterer, for the total years of InSAR analysis or for a number of scatterers near the mining area?

Response 8: Thanks a lot. We have revised the Table. The word “code” has been replaced by the word “Aera”.

Both the subsidence rate and cumulative subsidence columns are calculated for specific scatterers (block circles filled with blue color) shown in Figure 9(a) – (e).

Comments 9: In figure 14, could you please explain what the lines and dates represent?

Response 9: The lines represent the cumulative subsidence of 18 points (shown in Figure 12) along the railway. The dates, same with Figure 8, represent the time of SAR images.

The 2018-01-06 represents the first SAR image, and therefore the cumulative subsidence of all points is zero. The 2018-09-27 represents a SAR image acquired in Sept. 27, 2018, and the corresponding line represent the cumulative subsidence between Jan. 5 2018 and Sept. 27, 2018.

Round 2

Reviewer 4 Report

I would like thank the authors for addressing each of the comments. I have no further suggestions. I believe the paper is appropriate for publication.